# Stochastic Modelling to Assess External Environmental Drivers of Atlantic Chub Mackerel Population Dynamics

Ghoufrane Derhy [1], Diego Macías [2], Khalid Elkalay [1], Karima Khalil [1] and Margarita María Rincón [3,*]

1 Laboratory of Applied Sciences for the Environment and Sustainable Development, School of Technology, Cadi Ayyad University, Essaouira Al Jadida, Route d'Agadir, BP 383, Essaouira 44000, Morocco; ghoufrane.derhy@ced.uca.ma (G.D.); k.elkalay@uca.ma (K.E.); ka.khalil@uca.ma (K.K.)
2 Instituto de Ciencias Marinas de Andalucía (ICMAN), Consejo Superior de Investigaciones Científicas (CSIC), Campus Universitario Río San Pedro, Puerto Real, 11519 Cadiz, Spain; diego.macias@icman.csic.es
3 Centro Oceanográfico de Cádiz (COCAD-IEO), Consejo Superior de Investigaciones Científicas (CSIC), Puerto Pesquero, Muelle de Levante s/n, 11006 Cadiz, Spain
* Correspondence: margarita.rincon@ieo.csic.es

**Abstract:** The population dynamics of small and middle-sized pelagic fish are subject to considerable interannual and interdecadal fluctuations in response to fishing pressure and natural factors. However, the impact of environmental forcing on these stocks is not well documented. The Moroccan Atlantic coast is characterized by high environmental variability due to the upwelling phenomenon, resulting in a significant abundance and variation in the catches of small and middle-sized pelagic species. Therefore, understanding the evolution of stock abundance and its relationship with different oceanographic conditions is a key issue for fisheries management. However, because of the limited availability of independent-fishery data along the Moroccan Atlantic coast, there is a lack of knowledge about the population dynamics. The main objective of this study is to test the correlation between the environment conditions and the stock fluctuations trends estimated by a stock assessment model that does not need biological information on growth, reproduction, and length or age structure as input. To achieve this objective, the fishery dynamics are analyzed with a stochastic surplus production model able to assimilate data from surveys and landings for a biomass trend estimation. Then, in a second step, the model outputs are correlated with different environmental (physical and biogeochemical) variables in order to assess the influence of different environmental drivers on population dynamics. This two-step procedure is applied for chub mackerel along the Moroccan coast, where all these available datasets have not been used together before. The analysis performed showed that larger biomass estimates are linked with periods of lower salinity, higher chlorophyll, higher net primary production, higher nutrients, and lower subsurface oxygen, i.e., with an enhanced strength of the upwelling. In particular, acute anomalies of these environmental variables are observed in the southern part presumably corresponding to the wintering area of the species in the region. The results indicate that this is a powerful procedure, although with important limitations, to deepen our understanding of the spatiotemporal relationships between the population and the environment in this area. Moreover, once these relationships have been identified, they could be used to generate a mathematical relationship to simulate future population trends in diverse environmental scenarios.

**Keywords:** chub mackerel; population trend; SPiCT model; environmental covariates; Moroccan Atlantic coast

## 1. Introduction

Small and middle-sized pelagic fish are considered an essential resource in the marine ecosystem, which occupy an intermediate level in the trophic web [1]. In economic terms,

the small pelagic group is the main fish group harvested, representing approximately 39 million tons [2] and playing a crucial role in the ocean's food chain. Large fluctuations in the abundance of small and middle-sized pelagic fish have been observed over multiple decades and in very diverse areas, this instability being one of the main characteristics of these resources [3,4]. Their stock abundance can be driven by the main force of fishing mortality but can also be influenced by environmental variables. In this respect and based on biological characteristics, middle and small pelagic fish differ from each other [5]. The main differentiating characteristics between medium and small pelagic fish are mostly more plasticity in the food spectrum, a longer life span and major number of age groups, high mobility, and horizontal and vertical migration capacity [6]. However, middle-sized juvenile pelagic fish are in many aspects similar to smaller species. Due to their biological characteristics, middle-sized pelagic species tend to migrate more widely than smaller forms and are able to cross environmental barriers [5,6]. Considering all these characteristics, small and middle-sized pelagic species are typically very sensitive to environmental forcing [7–9].The processes explaining the impact of environmental covariates on small and middle-sized pelagic dynamics are not fully understood. According to Freon et al. (2005), their effects on small pelagic stocks, can be observed on three-time scales in which fisheries are influenced. The first category is short-term influences in which the environmental variability will affect only fish movement associated with changes in their aggregation pattern, which will affect the catchability of the harvested small pelagic species. The second type is medium-term influences, in which environmental covariates affect larval survival, which results in the interannual variability inthe recruitment success of the stock. The last category is long-term influences, which is related to the long-term fluctuations in the abundance trend of stocks, which are interpreted to be the result of environmental forcing.

The Moroccan Atlantic coast is one of the most productive fishing areas in the world because of the sustained Eastern Central Atlantic Upwelling system [10–12], which leads to a high biomass of small and middle-sized pelagic fish [13,14]. The small pelagic fishery occupies an important place in the Moroccan fisheries sector [15], with approximately 80% of the national fisheries production (National Fisheries Office (ONP), 2019). The volume of catches reached 1.4 million tons in 2019, an increase of 9% compared to the previous year [16]. This increase in catches requires intervention and analyses of the fishing-pressure level on the small pelagic stocks, as there are several biological signal indications of a decrease in the stock capacity within a highly unstable hydro-climatic environment [16]. Most studies in Morocco use fisheries and survey data to represent the abundance of stock and compare it with ocean dynamics observed by satellites [17–21].

However, based on existing data, including catches/landings and time series of available abundance indices provided by surveys, even with some gaps in the time series, exploration can be conducted in a data-poor context for stocks, in order to provide an estimation of the population dynamics trend using a Surplus Production model in Continuous Time (SPiCT, [22]). This approach is implemented for chub mackerel (*Scomber colias*) in the center and south of the Moroccan Atlantic coast (from Cape Cantin to Cape Blanc, Zones A, B, and C in Figure 1). Although this species is classified as a middle-sized pelagic fish, the stock is included in the Moroccan small pelagic fisheries. The stock corresponding to the center and south of the Moroccan Atlantic coast was selected as the study area, because it represents the largest proportion of the total small pelagic catches along the whole Moroccan Atlantic coast, 96% compared to 4% in the Mediterranean and the Northern Atlantic areas [16,23]. Chub mackerel population dynamics, stock identity, and status are still unknown in the whole area from the Moroccan coast to the southern Iberian waters [23]. The uncertainty about their population structure is mainly due to the lack of evidence of gene frequency differences among geographic samples; however, the biological traits suggest some similarity between Mediterranean and Atlantic individuals, but there are insufficient data and consistency to generate robust evidence on stock identity [23,24]. Despite this species occupying an important place in the commercial fishery of

the whole Moroccan coast, the available biological data (length-age composition, weight, maturity, etc) for this stock are very limited [25].

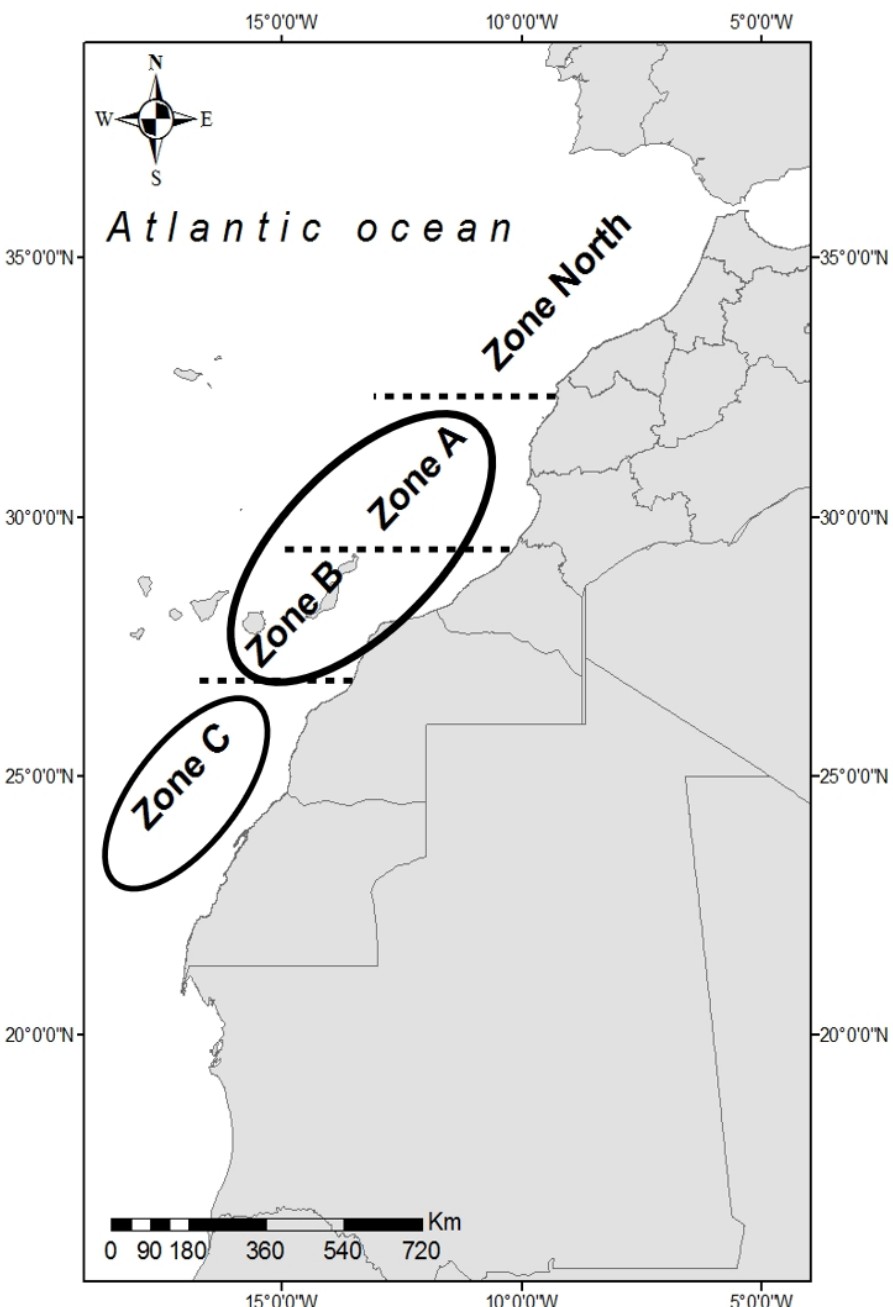

**Figure 1.** Chub mackerel study area along the Moroccan Atlantic Coast. Central area: A + B zones. Southern area: C zone.

Understanding the relationship between the environmental variations and the abundance of chub mackerel in the center and south of the Moroccan Atlantic coast is the main aim of this paper. First, a population trend is estimated based on four fisheries and surveys datasets available for the study area. The fishery dynamics are analyzed with a stochastic surplus production model able to assimilate data from surveys and landings for a biomass trend estimation including process and observation error [22]. Then, in a second step, the model outputs are correlated with different environmental (physical and biogeochemical) variables in order to assess the influence of different environmental drivers on population

dynamics. This two-step procedure is applied for the chub mackerel along the Moroccan coast, where all these available datasets have not been used together before. The results indicate that this is a powerful procedure, although with important limitations, that can deepen our understanding of the relationships between the population and the environment in this area. Moreover, once these relationships have been identified, they could be used to generate a mathematical relationship to simulate future population trends in diverse environmental scenarios.

## 2. Materials and Methods

### 2.1. Study Area

This study is focused on the central (between Cape Cantin and Cape Boujdor (32°32′24″ N–26°07′59″ N) and the south of the Moroccan Atlantic shelf (26° N-north Cap Blanc) (Figure 1). These two areas enclose the central and south stocks of small pelagic fish, mainly composed of sardines, anchovies, chub mackerel, horse mackerel, and occasionally sardinellas. These zones produce about 45% of the total catch of small pelagic fish in the Moroccan Atlantic area [15]. The abundance of these resources seems to be strongly correlated with the intensity, seasonal, and interannual variability of the coastal upwelling phenomena, which occur in the Grand Canary Ecosystem [26,27].

### 2.2. Biomass Trend Estimation

#### 2.2.1. Model Description

The SPiCT is a stochastic state-space model that provides stock status estimation and reproduces population dynamics by aggregating biomass across size and age groups by following Pella and Tomlinson equations [28]. The basic model equations are classified into process and observation equations.

The process equations describe population dynamics through exploitable biomass ($B_t$) and fishing mortality ($F_t$), while the observation equations link observed indices ($I_t$) and catches ($C_t$) with those dynamics as follows:

- **Biomass equation**

$$dB_t = rB_t\left(1 - \frac{B_t}{K}^{n-1}\right)dt - F_tB_tdt + \sigma_B B_t dW_t, \tag{1}$$

where $r$ and $K$ represent the intrinsic growth rate of the exploitable biomass $B_t$ and the carrying capacity, respectively, and $n$ determines the shape of the production curve. For the model implemented here, $n = 2$ was assumed imposing a Schaefer model on the population dynamics. The last term accounts for process noise, where $\sigma_B$ is the standard deviation of that process noise and $W_t$ is a Brownian motion, a Gaussian stochastic process that is frequently used when defining a stochastic perturbation of a measurement. Please note that the stochastic differential equation (SDE) described in (1) without the last term is an ordinary differential equation (more about SDEs in [29,30]).

- **Fishing mortality equation**

$$d\log(F_t) = \sigma_F dV_t \tag{2}$$

where $dV_t$ is a standard Brownian motion, and $\sigma_F$ is the standard deviation of the noise.

- **Index equation**

$$\log(I_t) = \log(qB_t) + e_t, e_t \sim N(0, \sigma_I^2) \tag{3}$$

where $q$ is a catchability parameter, and $\sigma_I$ is the standard deviation of the index observation error.

- **Catch equation**

$$log(C_t) = \log(\int_t^{t+\triangle} F_s B_s ds) + \epsilon_t, \epsilon_t \sim N(0, \sigma_C^2) \tag{4}$$

where $\sigma_C$ is the standard deviation of the catch observation error.

### 2.2.2. Model Data Input

The SPiCT model was fitted using landings time series from 1990 to 2018 and three different time series of abundance indices from acoustic surveys conducted by: the Norwegian research vessel RV Dr. Fridjof Nansen (1999–2015) using a pelagic trawl, the Russian RV Atlantida fleet (1994–2015) also using a pelagic trawl, and the Moroccan vessel R/V Al Amir Moulay Abdellah in the period 2000–2017 fished by purse-seine. It is important to consider here that the selectivity of the survey gears is an important factor mentioned in several papers. The availability of current data on the selectivity of commercial and survey trawls and purse seiners is critical to ensure that the assessments are based on the fishing gear characteristics actually used in the fisheries [31]. All these time series were obtained from the last CECAF report [32], and they are displayed in Figure 2.

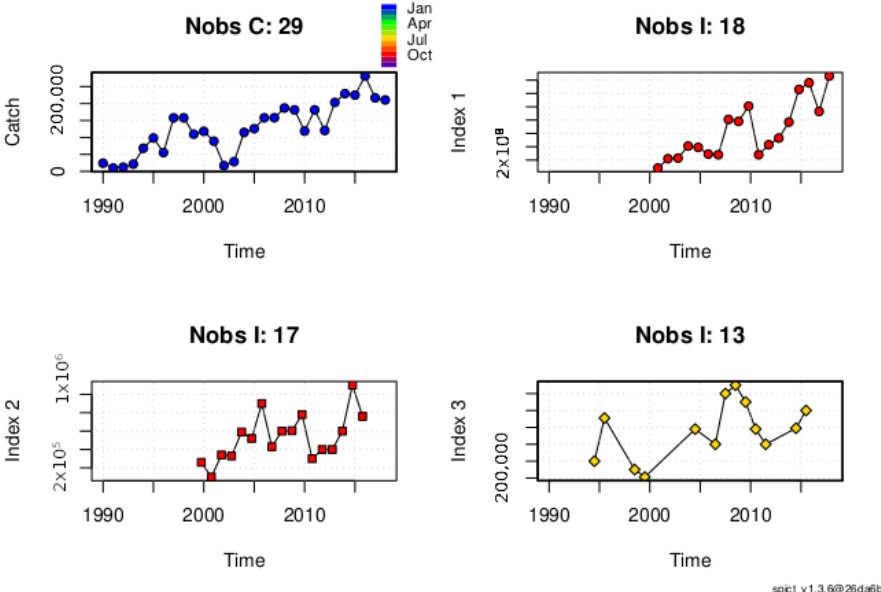

**Figure 2.** Data input summary. (**Top left**): chub mackerel catches in tons. (**Top right**): Amir Moulay Abdellah acoustic estimates (autumn). (**Bottom left**): Nansen acoustic estimates (autumn). (**Bottom right**): Atlantida acoustic estimates (summer).

### 2.2.3. Model Output Consistency Analysis and Model Implementation

The consistency for the model output was tested for the estimated relative biomass time series, which provides relevant information on population fluctuations. For this, a retrospective analysis (running the model by removing the last year of data at each iteration) was performed, and its corresponding measure of retrospectivity [33] was calculated basically as the difference between the most recent year's estimate, when removing one year or more, and the estimate for the same year provided by the model without removing any data. In addition, as the model assumed that one-step ahead (OSA) residuals of the fit were independent and normally distributed, bias, autocorrelation, and normality were tested for those residuals. Bias was determined by calculating the probability of the mean of residuals to be different from zero, autocorrelation by a Ljung-Box simultaneous test for 4 lags, and normality by a QQ-plot with a Shapiro test. The model and tests were implemented using the *SPiCT* R package (version 1.3.4, Vienna, Austria); more information on these tests and the model implementation is available at [22,34] (*Handbook for the SPiCT*).

*2.3. Comparison of Model Outputs with Physical and Biogeochemical Variables*

2.3.1. Sources of Environmental data

Environmental variables in the selected area of study (20° N to 35° N and −5° W to −20° W, Figure 1) were obtained from the Copernicus Marine data portal https://resources.marine.copernicus.eu/, accessed on 19 January 2021. Both physical variables and biogeochemical variables were downloaded using global reanalysis products that cover the region of interest and provide information at a monthly resolution for the entire time range considered in the stock model (1993 to 2018).

For physical conditions, the product 'GLOBAL_REANALYSIS_PHY_001_030' was used. This dataset contains 3D fields of the different variables at a spatial resolution of 0.086 degrees and monthly time-step. From the total dataset, the following variables were obtained:

- Bottom salinity;
- 3D temperature (°C);
- 3D salinity;
- 3D zonal velocity (m/s);
- 3D meridional velocity (m/s).

Biogeochemical conditions were downloaded from 'GLOBAL_REANALYSIS_BIO_001_029' product, which provides 3D fields at a spatial resolution of 0.25 degrees (i.e., coarser than the physical ones) at a monthly time-step. From this product, the following variables were downloaded:

- 3D chlorophyll ($mg/m^3$);
- 3D net primary production (NPP) ($mgC/m^3$);
- 3D oxygen ($O_2$) ($mmol/m^3$);
- 3D nitrate ($NO_3$) ($mmol/m^3$);
- 3D phosphate ($PO_4$) ($mmol/m^3$).

2.3.2. Comparison Procedure

The environmental data were spatially explicit and with monthly resolution, while the biomass estimates were merged for the whole Moroccan fishing grounds (except 'north'). Hence, in order to perform a correlation analysis, it was necessary to aggregate the environmental information. A direct approach was selected; thus, the monthly mean value for each variable integrated in the entire FAO fishing grounds (FAO 34.1.1 (Morocco Coast) and 34.1.3 (Sahara Coastal)) was computed. Two different integration depths were also tested: 50 m and 150 m, except for oxygen, was considered in the layer 100–200 m (thus providing a subsurface estimate of oxygen concentration).

This aggregation provided the monthly time series in the whole fishing area for the period 1993–2018 (the period covered by the stock estimates) for each integrated variable. From that 312 data point series, the values for each month of the year were extracted, creating 12 time series with 26 years (the first time series corresponded to all Januaries, the second to all Februaries, etc).

Further, we also computed the time series for each variable corresponding to each season, defined as: winter: December, January, and February; spring: March, April, and May; summer: June, July, and August; and fall: September, October, and November. Each one of those time series (monthly and seasonal) was compared with the biomass estimates provided by the SPiCT model by computing the Pearson correlation values with Matlab®.

The correlation between biomass estimates and environment was also analyzed to make use of the spatially explicit information provided by the hydrodynamic–biogeochemical models and computing the correlation coefficient (R) for each grid cell of the model domain.

However, and before performing the analysis of the relationship between environmental variables and the biomass estimates, a spectral analysis of the biomass time series was conducted in order to better understand its time dynamic and help assess the results of the correlation analysis.

Thus, Singular Spectrum Analysis (SSA, [35]) was applied to the 26 years of the relative biomass estimates. This analysis separates the independent signals (in frequency) that compose a given time series by isolating the different eigenvectors that compose the low-frequency signals within (e.g., Macías et al., 2014).

## 3. Results

### 3.1. Model Results

The estimated relative biomass and its retrospective analysis with their corresponding 95% confidence intervals (CIs) and with different scenarios shown by different colors are presented in Figures 3 and 4, respectively, while Figure 5 summarizes the test diagnostics for checking residual assumption violations. The estimated relative biomass time series is shown in Figure 3 (blue line). The $B_t/B_{MSY}$ time series showed three main peaks, the first in 1995, the second in 2008, and the third, which was the highest, in 2017. This last peak did not appear when running the model again removing the last year (red line in Figure 4), but in general the trend was very consistent until 2014, considering that the changes in the trend for recent years occurred due to the update of new information in the model.

The estimated relative quantity of stock biomass may have much less uncertainty and bias, although the noticeable uncertainty values observed in the peaks may be related to the lack of biomass index data in those peaks. In addition, the value of Mohn's $\rho$ for this variable was below 0.2 (Mohn's $\rho = 0.18$), in the range of an admissible retrospective pattern [36]. Figure 5 shows that there were not important model assumption violations except for the catch residuals, which were not normally distributed; this reflects some deficiencies in the model to fit the catch data but does not invalidate the model results.

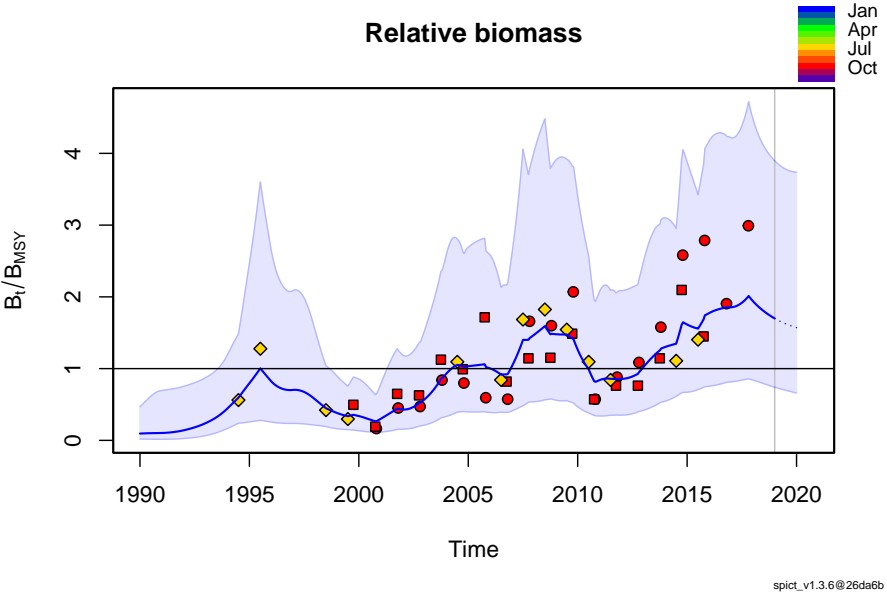

**Figure 3.** Atlantic chub mackerel. Estimated relative biomass time series (blue line) and estimated $B_{MSY}$ (black line). Data are shown using points colored by season; CI of $B_t/B_{MSY}$ estimates is respresented using dashed blue region.

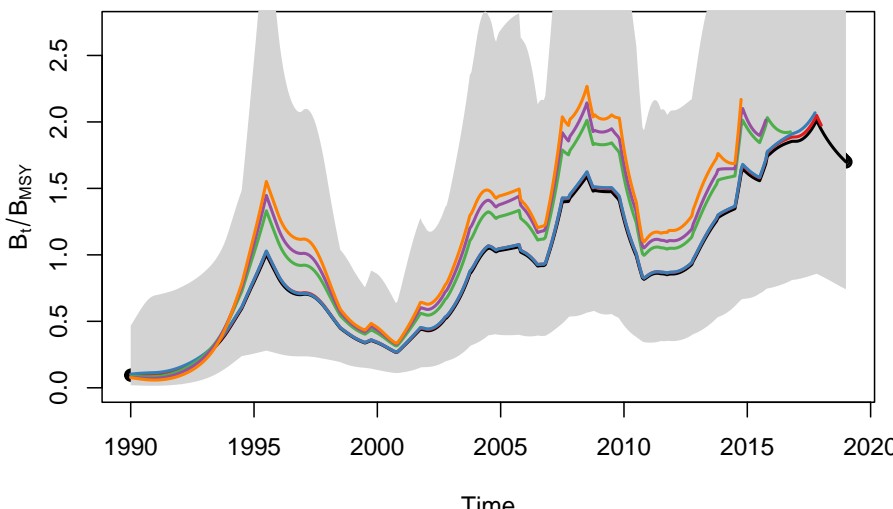

**Figure 4.** Atlantic chub mackerel. Retrospective analysis for biomass relative to $B_{MSY}$ from SPiCT assessment model. Different runs are shown by different colors of the same model.

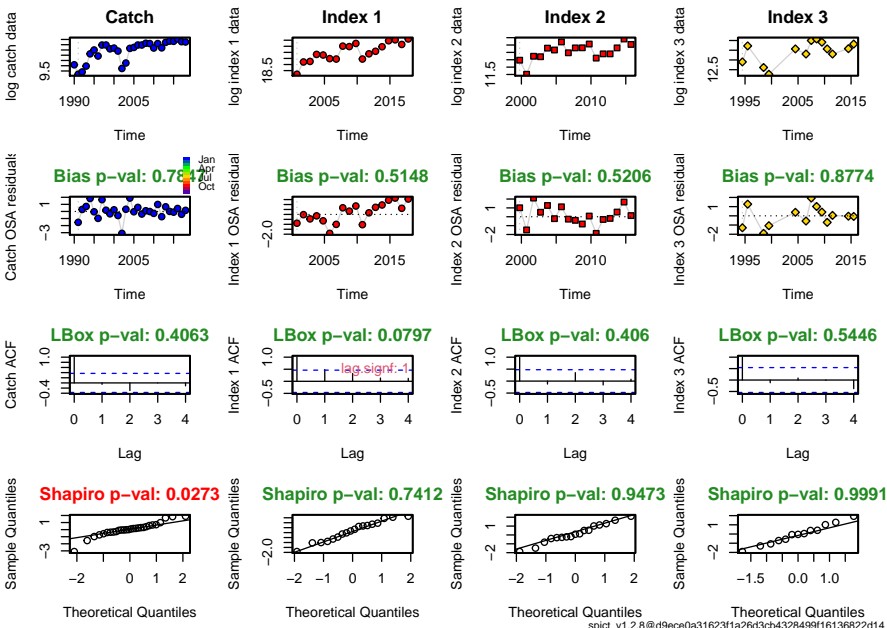

**Figure 5.** Atlantic chub mackerel. Summary diagnostics for violation of SPiCT model assumptions.

### 3.2. Spectral Analysis of the Relative Biomass Time Series

The two most important signals in the relative biomass time series detected by the SSA analysis were defined by eigenvectors 1 (first signal) and 2 and 3 (second signal) and represented 83% of the total variability (Figure 6, lower panel). The graphical representation of both low frequency signals (Figure 6, upper panel) revealed that the first signal (accounting for 46% of the variability) was a positive trend with no clear oscillations, while the second signal (representing 36% of the variability) was a 16.5-year oscillation with increasing amplitude (peaks in 1995, 2007, and 2017).

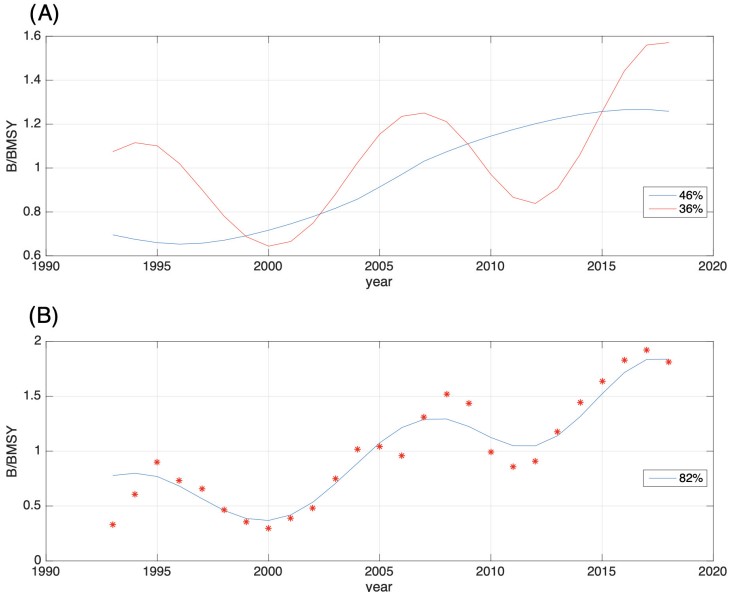

**Figure 6.** Atlantic chub mackerel. SSA analysis of the relative biomass time series. Upper panel, individual signals identified by the SSA analysis. Lower panel, original relative biomass series (red Asterisk)) and the reconstructed pattern using the two main signals (blue line) representing 83 percent of the total variability of the series (details in the text).

An important result of this analysis was the presence of the strong (46% of total variability) trend in the relative biomass time series. This indicates that detrending the time series for looking for correlations with the environmental variables (a common approach in this type of analysis) should be avoided. Furthermore, the second major signal indicated the presence of three high 'peaks' separated by two low 'valleys', even if the absolute values at the first peak were not higher than those on the second valley. All these characteristics of the biomass time series should, thus, be taken into consideration when performing the correlation analysis further on.

### 3.3. Correlation Analysis Biomass Trend–Environment

The monthly time series of environmental variables computed as described in Section 2.3.1 were used to search for linear correlations (Pearson correlation) with the relative biomass time series and to identify those statistically significant ($p < 0.01$), indicated with bold values in Table 1. The 3D zonal and meridional velocity and phosphate variables were not shown in the table, as these two variables were not correlated with the estimated trend in the relative stock biomass in all seasons.

As derived from the table above, the following variables with significant correlations and absolute R-values were selected for the analysis (marked in bold in the table):

- Mean 3D salinity (3D salt) in the upper 50 m (3D salt) in fall and winter ($R \sim -0.5$);
- Mean integrated chlorophyll (chlo) in the upper 150 m in all seasons except winter ($R \sim 0.6$);
- Mean integrated net primary production (NPP) in the upper 150 m in all seasons except winter ($R \sim 0.5$);
- Subsurface oxygen concentration (Oxy) (100–200 m) in all seasons ($R \sim -0.8$);
- Mean integrated nitrate (Nit) in the upper 150 m in all seasons ($R \sim 0.7$).

**Table 1.** Pearson correlation R values (by vertical integration) between environmental variables and Atlantic chub mackerel relative biomass time series. Bold numbers: significant correlation values ($p < 0.01$).

| Environmental Variables | SST | 3D Temp | 3D Salt | Chlo | NPP | Oxy | Nit |
|---|---|---|---|---|---|---|---|
| For 50 m integration depth | | | | | | | |
| Winter | −0.0001 | 0.009 | **−0.6** | 0.43 | 0.43 | **−0.83** | **0.38** |
| Spring | −0.05 | −0.10 | −0.38 | **0.61** | **0.57** | **−0.80** | **0.45** |
| Summer | 0.26 | 0.17 | −0.36 | **0.54** | **0.48** | **−0.65** | **0.43** |
| Fall | 0.31 | 0.22 | **−0.51** | **0.58** | **0.47** | **−0.73** | **0.43** |
| For 150 m integration depth (except for oxygen with 100–200 m) | | | | | | | |
| Winter | −0.00014 | −0.05 | **−0.49** | 0.36 | 0.34 | **−0.83** | **0.69** |
| Spring | −0.06 | −0.08 | −0.32 | **0.64** | **0.56** | **−0.80** | **0.68** |
| Summer | 0.27 | 0.06 | −0.16 | **0.58** | **0.48** | **−0.65** | **0.56** |
| Fall | 0.31 | −0.07 | **−0.42** | **0.64** | **0.44** | **−0.73** | **0.69** |

It is noteworthy to point out that the only non-biogeochemical variable with relation to the biomass time series was the 3D mean salinity in the upper 50 m of the region. All other considered physical variables showed non significant correlation values.

This first correlation analysis seemed to point out that larger biomass estimates were typically linked with an enhanced strength of the upwelling in the region, i.e., lower salinity, higher chlorophyll, higher net primary production, higher nutrients, and lower subsurface oxygen.

The selected environmental variables (bold values in Table 1) are displayed along the relative biomass time series in Figures 7–11 for the period 1993–2018.

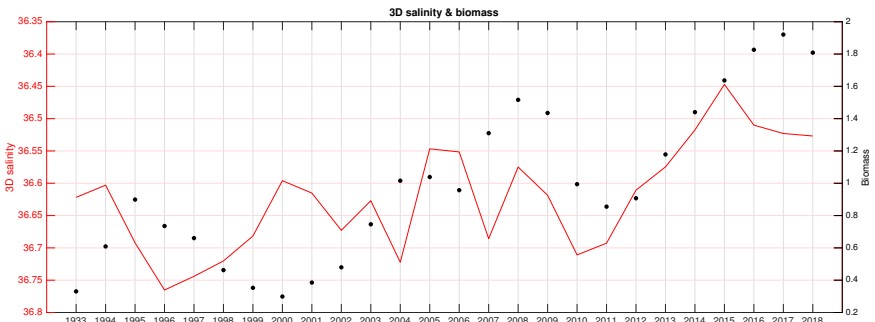

**Figure 7.** Atlantic chub mackerel. 3D salinity in fall (red line) and B/$B_{MSY}$ (black dots) for the period 1993–2018.

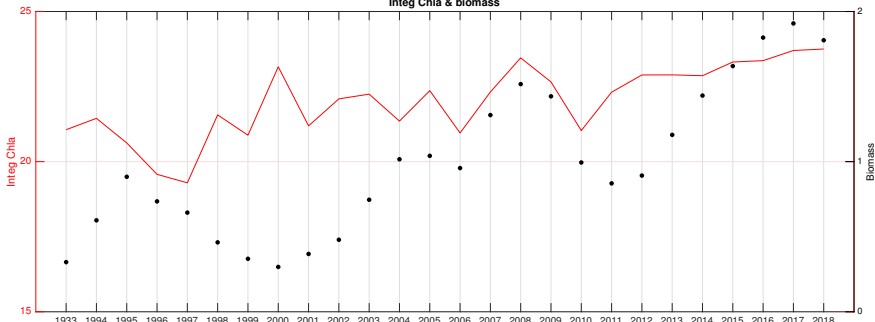

**Figure 8.** Atlantic chub mackerel. Integrated chlorophyll (0–150 m) in fall (red line) and B/$B_{MSY}$ (black dots) for the period 1993–2018.

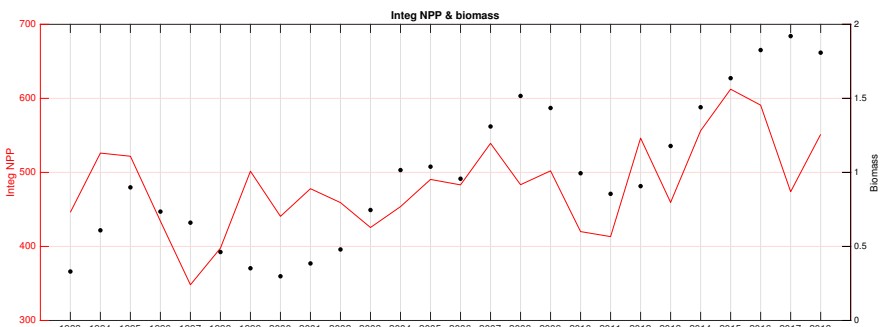

**Figure 9.** Atlantic chub mackerel. Mean integrated net primary production (0–150 m) in spring (red line) and B/$B_{MSY}$ (black dots) for the period 1993–2018.

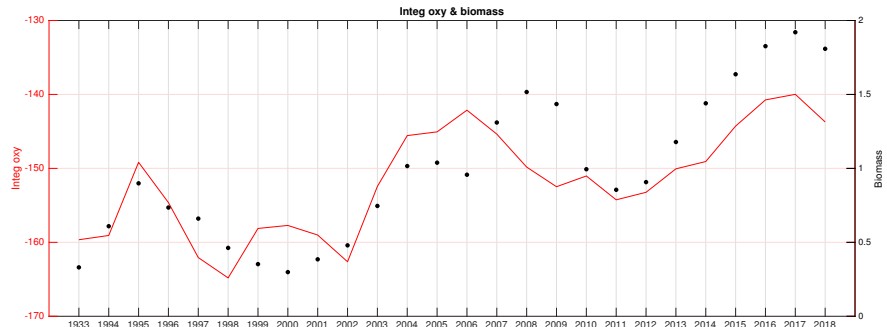

**Figure 10.** Atlantic chub mackerel. Integrated subsurface (100–200 m) mean oxygen concentration in fall (red line) and B/$B_{MSY}$ (black dots) for the period 1993–2018.

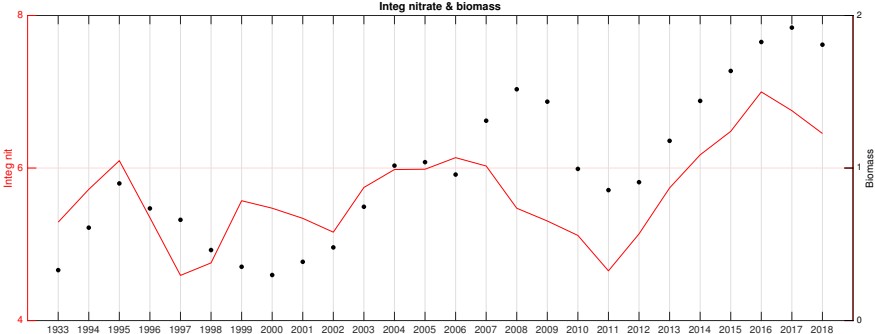

**Figure 11.** Atlantic chub mackerel. Integrated nitrate (0–150 m) in fall (red line) and B/$B_{MSY}$ (black dots) for the period 1993–2018.

Of all the tested variables, the higher correlation values corresponded to the subsurface oxygen and the integrated nitrate concentrations Figures 10 and 11. In both cases, high (above 0.7) and significant correlations were found for all seasons (Table 1) and for all individual months (not shown).

The spatial analysis of correlation between the biomass and environment is shown in Figure 12. The maximum values (in absolute terms) of R for the different variables were typically found in the region south of the Canary Islands and the nearby coastal African areas.

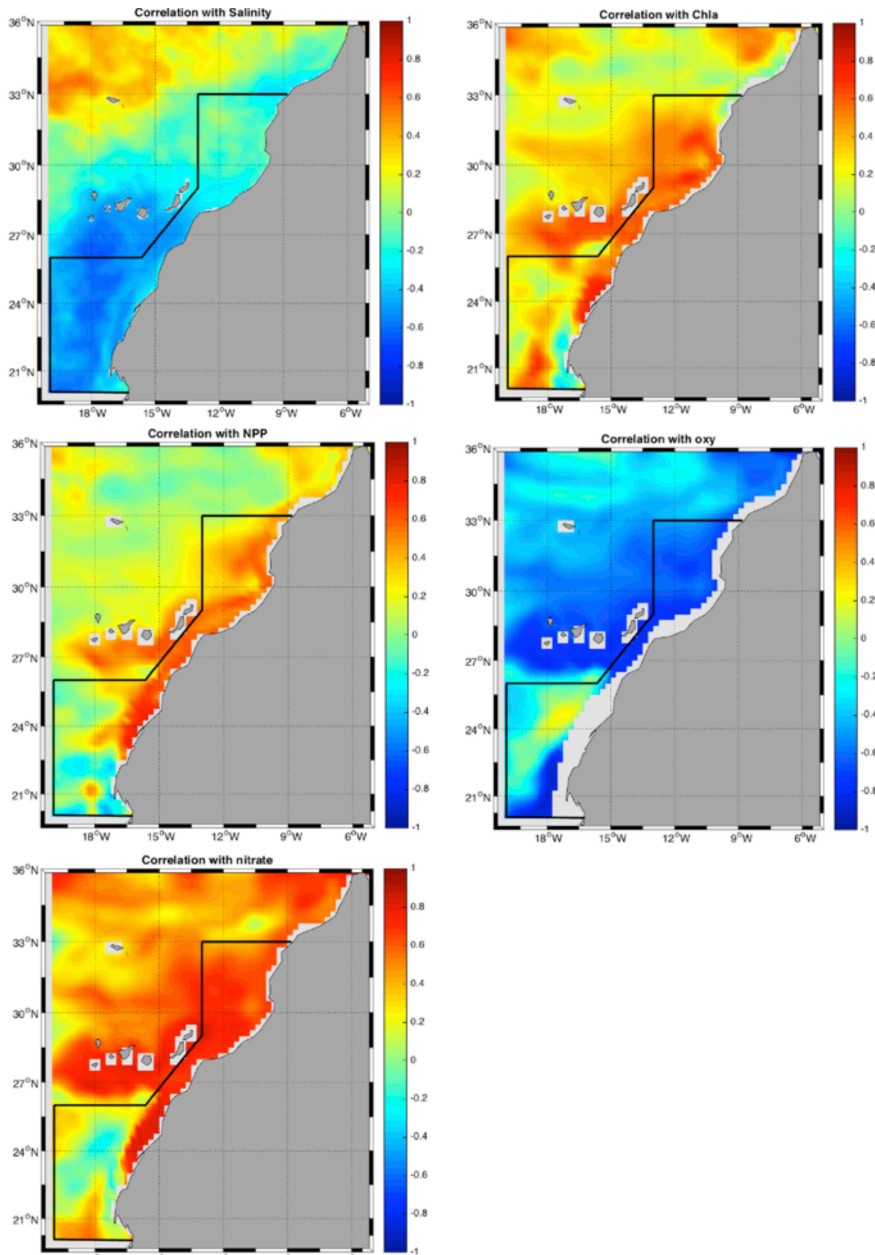

**Figure 12.** Atlantic chub mackerel. Spatially explicit analysis for the correlation between B/$B_{MSY}$ and the selected environmental variables (with significant correlations and absolute R–values (see Table 1)).

Alternatively, it is possible to determine which regions of the analyzed area presented the larger anomalies for the years in which the biomass estimates predicted larger-than-normal values. Figure 13 shows that the years with exceptionally large biomass estimates (over the percentile 75) corresponded to 2008, 2015, 2016, and 2017.

Figure 14 shows the anomalies (computed with respect to the climatological value) of the different environmental variables for those 'high biomass' years. This analysis corroborates that those years with a particularly high biomass of c hub m ackerel in the region correspond to years with enhanced strength of the upwelling (lower salinity, higher chlorophyll, higher net primary production, higher nutrients, and lower oxygen), with the most acute anomalies normally associated with the southern part of the studied area (as already indicated by the spatially-explicit correlation analysis above).

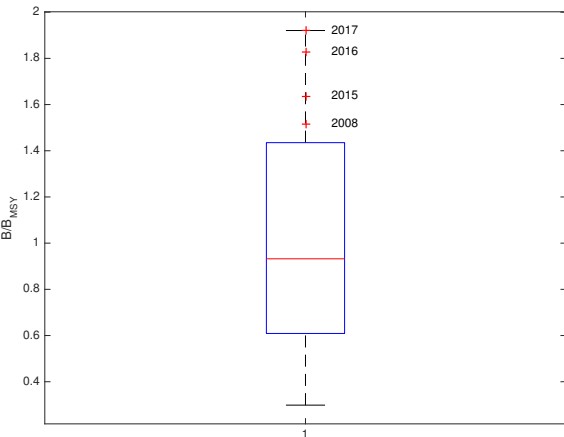

**Figure 13.** Atlantic chub mackerel. Boxplot of B/$B_{MSY}$ time series. The red line marks the median, the blue lines the 75 percentile range, and the dotted line the interval of confidence.

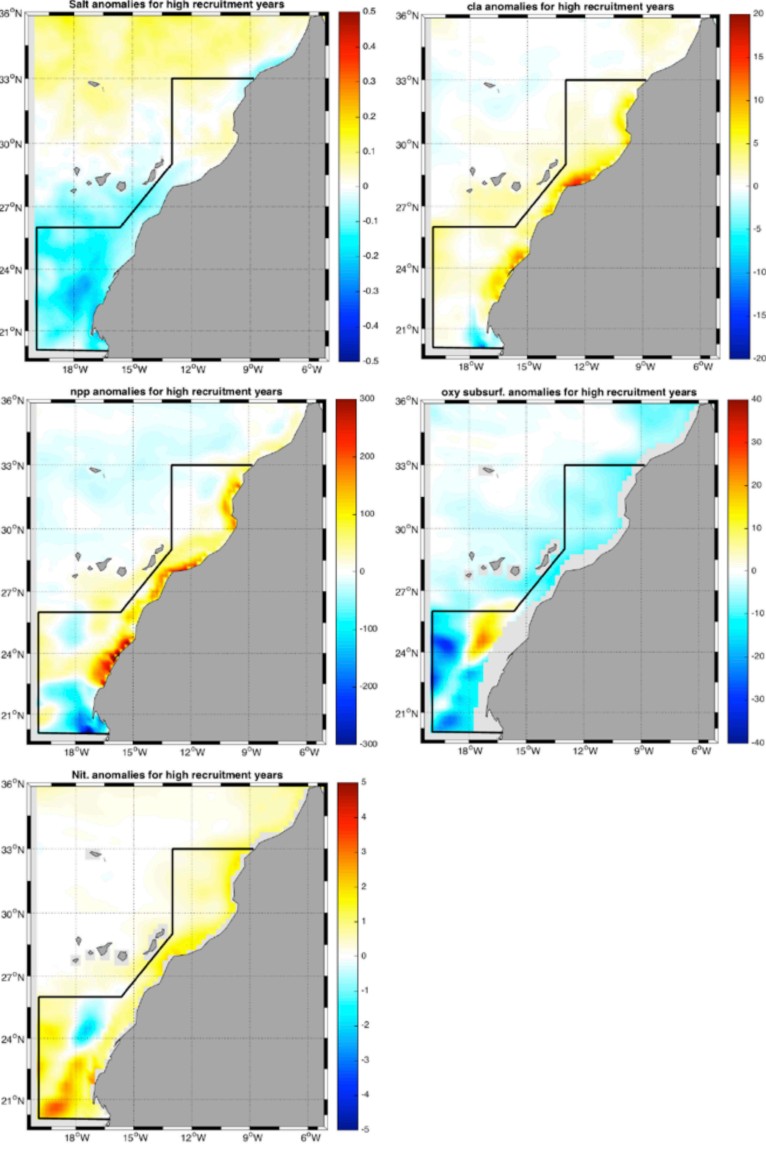

**Figure 14.** Atlantic chub mackerel. Anomalies for the different environmental variables in years of high B/$B_{MSY}$ time series values (2008, 2015, 2016, and 2017).

## 4. Discussion

### 4.1. Methodological Considerations

The effect of the environmental variations on the distribution and abundance of chub mackerel is an important issue that continues challenging scientists and fishery managers. The main limitation on determining the relationships between the environment and population dynamics is the short datasets that are generally available [37,38]. In addition, the stock is also harvested in Morocco by artisanal fleets, which are characterized by a partial or total lack of data on landings and effort; therefore, it is difficult to estimate the real importance of artisanal fisheries in the stock analysis [24]. Given the unavailability of fishery-independent information, a reliable estimate of the historical biomass series is not easily obtained. The procedure presented here combined all the fishery and surveys data sets available in the region to generate an abundance trend for chub mackerel. This trend could be the first step in determining the main environmental factors influencing population dynamics.

This abundance time series was estimated using a SPiCT model, a model that has been used for scientific advice in the ICES framework by providing trends (category 3 and 4 stocks [39]) but also by providing quantitative assessments (category 1 stocks [40]). One of the main advantages of this model is that it is based on a continuous-time formulation as opposed to fixed and constant time steps allowing incorporation of arbitrarily sampled data [22,40]. It is therefore easier to fit this model to a mixture of annual, semiannual, and quarterly data [41]. In addition, the model is fully stochastic in the sense that the observational error is included in the catch and index observations, and the process error is included in the fishery and stock dynamics [22].

The main output of the SPiCT model used in this study was the estimated time series of relative biomass ($B_t/B_{MSY}$), which is less sensitive to the choice of biomass representative of the stock [22] and, in fact, is considered a robust estimator of population dynamics [42]. The uncertainty in the relative biomass time series estimates was much lower by assuming a Schaefer-type production curve. The consistency of the key model outputs was evaluated performing a retrospective analysis (e.g., [33,43]) that resulted in an acceptable Mohn's rho value. Despite the model's success in estimating stock abundance by integrating all available data from different time series and data sources, it did not include environmental variability. This unmodeled process variability was included as a random error (process error) term in the biomass dynamic equation [22]. It has been assumed that the environmental changes can affect both the carrying capacity and the rate of stock production [44,45].

Many studies have identified possible assumptions that link environmental effects to changes in stock abundance [46–48], Rincón et al. [49], so it should be required to include environmental variability in fisheries' management approaches [50]. However, it is necessary to first understand the relationship between the environment and fluctuations in stock biomass and to identify the specific factors that may impact stock abundance in order to include these external factors in the fisheries' management approaches [4].

### 4.2. Moroccan Chub Mackerel Fishery

In the central zone, chub mackerel is exploited by the Moroccan coastal purse seiners, which mainly target sardine, making catches of chub mackerel in this area less variable [32]. In the southern zone of Morocco, this species is fished by Moroccan coastal purse seiners as well as by Moroccan trawlers such as Refrigerated Sea Water (RSW), and the catches have fluctuated over time due to the presence and absence of Russian and European pelagic trawlers that continued to fish under the Morocco–Russia and Morocco–EU bilateral fisheries agreements in certain years [23,32,51]. For most fisheries, changes in the time on fishing gear, spatial distribution of the harvested fish population, and species targeting will strongly determine apparent trends in population abundance [52]. The fishery–based approach is not sufficient to indicate the stock status, suggesting that other unbiased information should be added, which may also control stock dynamics [53]. A previous study found that this information included a marginal 2% improvement in the success rate

of describing population dynamics [52]. In our case, the fishing conditions have remained more or less constant during the last five years; hence, the fluctuations in the population should be driven by other factors.

The exploited stock of chub mackerel is harvested in the upwelling ecosystems of the Atlantic coast of Morocco, which presents a high latitudinal and temporal variability inen vironmental conditions. According to previous studies, in addition to exploitation, the causes of large variations in the small and middle-sized pelagic fish biomass are related to natural variability, mainly environmental changes [3,54]. However, factors that affect recruitment of the small pelagic fishes might be substantially different from those affecting the middle-sized ones [6].

### 4.3. Environmental Effect on Chub Mackerel Abundance Variability

This work identifies environmental factors related to the abundance of chub mackerel. Among the environmental factors studied, salinity, chlorophyll concentration, net primary production, oxygen concentration, and nitrate concentration were the main parameters significantly correlated with the spatiotemporal variations of chub mackerel, which are also factors linked to the upwelling intensity. Salinity showed a negative relationship with the estimated relative biomass, which is consistent with chub mackerel behavior: this species aggregations tend to be higher in shallower areas characterized by lower salinity; the negative relationship appears to be related to the spawning success [55], which suggests a link between salinity fluctuations and the stock spawning. This area is characterized by a high spatial variability in primary productivity, which is mainly due to strong and sustained coastal wind stress, resulting in offshore Ekman transport and upwelling off nutrient- rich subsurface water [4]. It has been shown that primary productivity partly controls the abundance levels of exploited small pelagic stocks [56], which was confirmed by the strong linear relationship found between the net primary production and the chub mackerel abundance. However, the feeding strategy of middle-sized pelagic fish differs considerably from that of small pelagic [6]. Several studies have identified appropriate habitat characteristics for spawning and larval growth of chub mackerel with an increase in primary production [57,58]. The waters of the northwest African coast contain high values of inorganic nutrients such as nitrate, which has a positive effect on phytoplankton growth [59]; this may impact the abundance of zooplankton (large copepods, mesozooplankton) in particular, which are major contributors to the diet of middle pelagic fish and are constantly highly abundant [6] and may also indirectly impact the chub mackerel biomass. Once phytoplankton cells die or are eaten and excreted by zooplankton, the cells sink and are remineralized by bacteria. This process occurs in the subsurface layer and reduces the concentration of dissolved oxygen, which is usually the case in strong upwelling [7]. In our analysis, the dissolved oxygen concentration showed a significant ($p < 0.001$) negative correlation with the estimated chub mackerel relative biomass, a parameter that is related to the upwelling intensity, as explained above.

These correlation analysis results allowed us to propose that fluctuations in the environment play a major role in the abundance of the chub mackerel stock, which showed that the abundance of this harvested stock population continued to increase until 2017; then, a slight decrease was observed in 2018, indicating that the evolution of this stock can follow changes in the environmental factors mainly related to the upwelling intensity that also changed during those years [60].

In our study region, the upwelling is an Ekman-type, where new nutrient inputs and turbulence are related to wind speed [38]. Cury and Roy (1989) showed that small-fish productivity can be sometimes positively and other times negatively correlated with the upwelling intensity [7,61–64]. In an Ekman type of upwelling, the annual recruitment increases with upwelling intensity until the wind speed attains a value of about 5–6 m.s-1 and declines for higher values of wind (strong upwelling), even if the primary production increases [7]. This suggests that the relationship between recruitment variability and annual upwelling indices are dome-shaped in Ekman- type upwellings and linear for non-Ekman-

type upwellings [7,61]. This may explain the strong fluctuations observed in the estimated stock abundance trend, despite the stability of the catches.

It has been hypothesized in Martins et al. (2013) that the Portugal–Cadiz waters provide a nursery for a large population spawning north or more probably in Moroccan waters, representing the core of both chub mackerel biomass and catches of the northwest Atlantic [65]. This movement requires particular physical mechanisms to transport early life stages from the spawning areas to the nursery habitat [66]. The size and age composition of landings show the presence of an ascending size/age gradient from the Portugal–Cadiz area to the Moroccan waters [66]. In the northern area of Morocco, the small sizes (juveniles/young adults) are dominant [67,68], and the same age and length stock distribution was observed in the Portugal-Cadiz area. In contrast, most individuals landed in the southern zone are large (in the range of 30cm) and older (6+ age) [65,67]. This confirms the migration of juveniles and young adults individuals from the northern part of Morocco to the south as they become older. Before each spawning season, chub mackerel grow and accumulate energy during the wintering period. Wang et al. (2021) have reported that for the chub mackerel stock in the northwest Pacific Ocean, the environmental conditions in the wintering area have a strong impact on the chub mackerel abundance. In our case, we can say that if the wintering area is the southern part of the study region, this confirms our results of the spatial correlation analysis (Figure 12) showing that the high values of the absolute R-values between the estimated abundance and environment were detected in the southern region. In addition, the spatial distribution analysis of anomalies (Figure 14) illustrates that the most acute anomalies of the environmental variables corresponding to the years with strong upwelling intensity and therefore the highest stock biomass were typically observed in the southern part of the study area, which may explain the increase in catches reported especially in the south.

### *4.4. Summary*

Based on our results, it can be said that the chub mackerel population in this study area shows fluctuations due to the upwelling variability, where we were able to identify the factors related to the upwelling intensity, which can influence the dynamics of this stock. In the most common situation, environmental variability impacts the distribution of fish, which may be associated with changes in the aggregation pattern and thus affects the catches and availability of harvested stocks [9]. Therefore, stock assessment models need to be adapted to consider both the fishery impact and the environmental effects [45]. A similar relationship was tested for the white shrimp stock in Senegal using Fox and Freon surplus production models including environmental effects (the upwelling intensity and the primary production), to analyze the fluctuations of the stocks in a long time series [45]. Comparing the results, their simple model explained just 17.7% of the observed abundance variance, and the model including the environmental effect of upwelling explained 64.9% of this variance [45], showing the significant effect of environmental variability in the stock assessment model results.

The results obtained during this work provided additional knowledge on the chub mackerel population dynamics along the central and southern Atlantic coast of Morocco using a new approach. To improve this approach, we are interested in developing a stock assessment model (with an updated analysis using all the available information at that time) using a mathematical relationship between the stock abundance and environmental covariates. This mathematical relationship will be a tool to simulate scenarios on population dynamics under different environmental conditions and will provide a significantly better explanation of the variance observed for the stocks in which fishing effort alone is not a sufficient parameter to analyze changes in the stock abundance time series.

**Author Contributions:** G.D., M.M.R. and D.M. contributed to the design and implementation of the research, to the analysis of the results, and to writing of the manuscript. K.K. and K.E. contributed to the interpretation of the results. All authors reviewed the results and approved the final version of the manuscript.

**Funding:** This study was carried out within the Farfish and EuroSea projects. The FarFish project has received funding from the European Union's Horizon 2020 research and innovation programme under grant agreement No 727891. The EuroSea project has received funding from the European Union's Horizon 2020 research and innovation programme under grant agreement No 862626. However, the paper does not necessarily reflect European Commission views and in no way anticipates the Commission's future policy in the area.

**Institutional Review Board Statement:** Not applicable

**Informed Consent Statement:** Not applicable

**Data Availability Statement:** Not applicable

**Conflicts of Interest:** The authors declare no conflict of interest. The funding sponsors had no role in the design of the study; in the collection, analyses, or interpretation of data; in the writing of the manuscript, and in the decision to publish the results.

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
