# Peer review of "Stochastic Modelling to Assess External Environmental Drivers of Atlantic Chub Mackerel Population Dynamics"

_sustainability, doi:10.3390/su14159211_

Round 1

Reviewer 1 Report

This MS employs two-steps procedure modeling the correlations between the Atlantic chub mackerel population dynamics and environmental variables, without biological data. The results explain this stock’s spatio-temporal trends with key environmental conditions, which could also be used to simulation analysis. It’s well written and the analysis is good. 

1, ABSTRACT, it may be better to write less background, and more results. In line 9, “in this region” may be better changed to “in Moroccan Atlantic coast”.

2, KEYWORDS, it may be better to add “Moroccan Atlantic coast”

3, Line 150-151, “FAO 34.1.1 and 34.1.3” are not found in Figure 1.

4, Figure 3/4, symbols and lines are not explained.

5, In Line 202, Figure 12, and Line 332, it should be expressed as “the absolute R-values”.

6, DISCUSSION, it may be better to add subtitles.

7, References 29/37 are incomplete. 

Reviewer 2 Report

Comments to the manuscript by G. Derhy et al. Code: 1769683

The paper presents from a methodological point of view a sound and largely innovative proposal to include environmental information in stock assessments of pelagic fish stocks under exploitation, specifically Atlantic chub mackerel in Moroccan Atlantic waters. The incorporation of environmental information allows a better understanding of the population dynamics of Atlantic chub mackerel and therefore establishes a framework for developing better predictions on the response of the resource to fishing activity and environmental changes. The authors also succeed in the methodological approach with their two-step model to analyze both the population dynamics of Atlantic chub mackerel and the time series of environmental variables. They incorporate stochastic models where process and observation errors are included and therefore there is a good treatment of the associated uncertainty. In summary, the work shows well defined objectives and a very clear and appropriate methodological approach with which interesting results have been obtained. I would like to congratulate the authors for the clarity and argumentative logic shown throughout the manuscript. The subject of the article, which has to do as an ultimate goal with the sustainability of Atlantic chub mackerel fisheries and as a methodological example for other fisheries, falls within the editorial line of the journal Sustainability.

This positive assessment notwithstanding, the manuscript suffers from some weaknesses precisely in the specific aspect of the assessment of Atlantic chub mackerel. The authors should also better specify the general biological framework in which they have framed the Atlantic chub mackerel, as well as correct a number of other inaccuracies throughout the text.

GENERAL COMMENTS:

The authors include Atlantic chub mackerel as a small pelagic fish. It is true that in several literature sources, mackerels, jacks, horse mackerels and other pelagic fishes of similar size are considered small pelagic fishes along with sardines, anchovies and sprats. However, depending on the objective pursued, this generalization is not always useful or realistic. Mackerels and horse mackerels belong to another size category, the so-called middle-sized pelagic fish. These middle-sized pelagic fish have life history characteristics that are different from the small pelagic fish. One of the most important is longevity. Other important characteristics are diet, their role in the food chain, migratory capacity, lower natural mortality and greater population stability in the face of environmental and anthropogenic impacts, largely due to greater longevity and lower natural mortality. It is true that despite these differences, there are also points in common such as the variability of the populations and the influence of environmental variables on their behavior and dynamics, although to a lesser degree than in small pelagics.  Therefore, the authors should take this into account throughout the text and treat mackerel as a middle-sized pelagic fish precisely and not as a small pelagic fish along with sardines and anchovies.

Regarding the Atlantic chub mackerel assessment, the uncertainty about the stock definition has to be included in the text. Pragmatically, zones A, B and C of the Moroccan coast have been considered as the stock area. But why not also consider the northern zone? This has to be justified and if it cannot be done, it has to be explained that a number of assumptions have been made here but that there are serious uncertainties.

The model also states that three series of data from fishery-independent research surveys are used. In this case it is necessary to explain what type of surveys they are and what type of fishing gear they use. Depending on the type of gear the representativeness for Atlantic chub mackerel will be better or worse. This should be explained in the text.

The article actually presents a framework or scheme to evaluate data-poor stocks and how to include environmental variables in the analysis. The example of Atlantic chub mackerel from the Moroccan coast is given as an example and this is all well and good. But the paper does not present current information and therefore an assessment for the present management of Atlantic chub mackerel, as the historical data series is up to the year 2018, somewhat far from the present. The authors should acknowledge in the paper this limitation.

SPECIFIC COMMENTS:

- Throughout the text, wherever it says "small pelagics" it should read "small and middle-sized pelagic fish" or sometimes "middle-sized pelagic fish".

- Lines 11-12.  "... stock assessment model that do not need biological information as input". This sentence is not entirely correct. The stock unit must always be taken into account in the model and this is biological information. Please change the wording for example specifying "... stock assessment model that do not need biological information on growth, reproduction and length or age structure, as input.."

- Line 22. I see better "Population dynamics" instead of "Population trends".

Introduction:

- Lines 30-31. Include here the biological characteristics of middle-sized pelagic fish compared to small pelagic fish.

- Line 39: Delete: "..., which is related to long term influences,..."

- Line 47. Change to: "This increase in catches require intervention...". We do not know if there has been an increase in fishing mortality. This requires an assessment. What we have for sure is the increase in catches. If the stock has increased a lot the increase in catches may not mean an increase in fishing mortality.

- Line 50: Delete the word "only". 

- Line 57: Explain what the basis for considering this stock unit is and include existing uncertainties.

Material and Methods:

Lines 78 and 79: Coordinates are misspelled. It puts degrees instead of seconds. Please correct.

Line 80: Take into account what has been said about stock identification.

Lines 98 and 100: Explain what is "Brownian motion".

Line 151. Change: "...were computed...".  

Legend to Figure 3. Change to: "Atlantic chub mackerel. Time series of Biomass relative to BMSY from SPiCT assessment model".

Results:

Lines 172-182: In the results there is also something to be said about the value of the uncertainty, especially noticeable in the relative biomass peaks.

Legend to Figure 4. Change to: "Atlantic chub mackerel. Retrospective analysis for Biomass relative to BMSY from SPiCT assessment model.

Legend to Figure 5. Change to: "Atlantic chub mackerel. Summary diagnostics for violation of SPiCT model assumptions."

Legend to Figure 6. Add at the beginning: "Atlantic chub mackerel."

Legend to Table 1. "Pearson correlation R values (by vertical integration) between environmental variables and Atlantic chub mackerel relative biomass time series". 

Table 1. Explain what has happened to the variables not covered in Table 1: 3D Phosphate, and 3D zonal and meridional velocity.

Legends of figures 7, 8, 9, 10, 10, 11, 12, 13 and 14. Add at the beginning: Atlantic chub mackerel.

Lines 220-224. The spatial approach should also be explained in the material and methods section.

Discussion:

Lines 237-239: This is an important limitation but it should also be noted that most of the stocks are data poor stocks where there are difficulties in catch and effort data or no fishery independent information, etc. Therefore, getting a reliable estimate of the historical biomass series is difficult.

Lines 250-258. Include something about the uncertainty in the estimate of relative biomass shown by the model....

Lines 277 - 280. Take into account that we are dealing with a middle-sized pelagic fish whose dynamics are different from the small pelagics: anchovies, sardines and sprat.

Lines 291-302. It is necessary to be more precise here and include references on the diet of Atlantic chub mackerel, mainly zooplanktivorous and with a percentage of ichthyophagous in the larger specimens. It is not exactly the same as the small pelagics sardines, anchovies etc. However, the upwelling conditions will also favor zooplankton production in the second instance and therefore benefit the Atlantic chub mackerel.
